# A 3D Space-Time Non-Local Mean Filter (NLMF) for Land Changes Retrieval with Synthetic Aperture Radar Images

Antonio Pepe

Institute for Electromagnetic Sensing of the Environment (IREA), Italian National Research Council, 328, Diocleziano, 80124 Naples, Italy; pepe.a@irea.cnr.it; Tel.: +39-0817620617

**Abstract:** Sequences of multi-temporal synthetic aperture radar (SAR) images are routinely used for land-use land-change (LULC) applications, allowing the retrieval of accurate and up-to-date information on the state of the Earth's surface and its temporal variations. Change detection (CD) methods that rely on the exploitation of SAR data are, generally, made of three distinctive steps: (1) pre-processing of the SAR images; (2) comparison of the pairs of SAR images; and (3) the automatic extraction of the "changed areas", employing proper thresholding algorithms. Within this general framework, the reduction in speckle noise effects, which can be obtained by applying spatial multi-looking operations and ad hoc noise filters, is fundamental for the better detecting and classifying of changed regions. Usually, speckle noise filters are singularly and independently applied to every SAR image without the consideration of their inherent temporal relationships. In particular, most use local (spatial) approaches based on determining and averaging SAR backscattered signals related to neighboring SAR pixels. In this work, conversely, we explore the potential of a joint 3D space-time non-local mean filter (NLMF), which relies on the discrimination of similar features in a block of non-local SAR pixels extracted from the same or different SAR images. The theory behind non-local-mean filters is, first, shortly revised. Then, the developed space-time NLMF is applied to a real test case for the purposes of identifying flooded zones due to the massive inundations that hit the Kerala region, India, during the summer of 2018. To this aim, a set of 18 descending SAR images collected from the European (EU) Copernicus Sentinel-1 (S-1) sensor was exploited. The performance of the developed NLMF has also been assessed. It is worth remarking that the proposed method can be applied for the purposes of analyzing a heterogenous set of natural and/or artificial disastrous conditions. Further, it can also be helpful during the pre-processing stages of the sequences of SAR images for the purposes of CD applications.

**Keywords:** synthetic aperture radar (SAR); non-local-mean-filter (NLMF); change detection (CD); flooding

## 1. Introduction

The last two decades have represented for the remote-sensing (RS) community a fructuous period characterized by the accelerated exploitation of synthetic aperture radar (SAR) methodologies for Earth's surface monitoring [1–13]. In this context, a series of regularly acquired SAR data are typically used to track land changes at different spatial and temporal scales, using both interferometric SAR (InSAR) [14–18] and amplitude-based change detection (CD) approaches [19–25]. More specifically, over the recent eight years, a new stimulus for developing and applying new SAR-driven techniques has been delivered by the possibility of accessing extensive archives of SAR images collected by the European (EU) Copernicus mission through the twin Sentinel-1A/B radar instruments. The free and open access policy of Sentinel-1 [14,26–31] and the weekly repetition frequency of the SAR observations have been fostering the evolution of novel SAR/InSAR methods, thereby allowing suitable and up-to-date observations of the Earth's surface. However, innovative technological developments have resulted in fundamental responses for the purposes of the adapting of existing processing codes, as well as of the handling of interferometric wide

(IW) swath S-1 SAR data [32–39]. The IW SAR images are collected through the terrain observation with progressive scans (TOPS) mode [39], which is the principal acquisition mode of S-1 over lands. InSAR and CD methods can jointly be used to recover damage maps on a global scale after a natural or human-made hazard occurs. Many features can be extracted from the sets of SAR images; the most used factor for RS analyses is the radar backscattering coefficient, which is an essential indicator of ground conditions due to its sensitivity to surface roughness. Moreover, an alternative to the exclusive use of SAR backscattering signatures for the generation of land-use land-change (LULC) products [40–44] is found in the exploitation of interferometric SAR approaches, such as those based on tracking the InSAR coherence changes [45–51].

The performance of InSAR and CD methods are impaired by the level of noise that affects the used set of SAR images. Several approaches have been developed to discriminate the valuable signals from the noise and filter out artefacts from SAR/InSAR products [52–64]. Most operate independently on every SAR image product (i.e., a single SAR image or single SAR interferogram) by exploiting exclusively spatial information. For instance, multi-look processes [65–68] are typically used in order to reduce the variance of the speckle from amplitude SAR images by averaging local signals while degrading the spatial resolution of the filtered SAR images. Ad hoc (local) de-speckling and noise filtering methods operating on SAR interferograms have been developed and widely used by the scientific community [63,69–73]. It is worth noting that the role of SAR de-speckling methods, implemented explicitly for change detection purposes, has been widely addressed in the literature. Interested readers can refer to high-impact papers in [8,24,74–80] to discuss this interesting topic, comprehensively. Of course, on the one hand, de-noised SAR images emphasize significant changes and, on the other hand, tend to destroy those minor changes that are within the error bar of the noise artefacts.

Recently, non-local techniques have also been developed [2,81–83]. The key idea behind these more recent methods is that there are enough pixels over a scene that, though not necessarily located in nearby areas, could be used to suppress the noise while preserving the spatial resolution of the original (unfiltered) images, with a significant impact on the capability to discriminate features at finer spatial scales (even at the pixel-scale level). The original method proposed by Buades [83] was developed in order to discriminate and filter out noise signals that can be represented as additive Gaussian processes. The adaptations of NLMF methods to work with amplitude SAR signals, which are affected by multiplicative speckle noise, have subsequently been addressed in several papers (e.g., [64,84–90]). In particular, the extension of 2D (space) NLMF filters for de-speckling sequences of 3D multi-temporal SAR images was also recently proposed [91–94]. Other than the peculiarities of every single method, these approaches all conduct non-local mean filters in the separate spatial and time domains, using either the original or log-ratio SAR images.

In this work, we propose and analyze a non-local mean filter (NLMF) approach applied to the sequences of 3D multi-temporal SAR images, thus extending the use of NLMF to the space-time domain. Specifically, non-local information associated with pixels belonging to different multi-temporal SAR images is extracted and averaged in order to improve the NLMF performances further and to produce adaptive temporal multi-looking products for CD analyses. Moreover, the presented method can also be adapted to work with InSAR data. It will be shown how NLMF impacts the retrieval procedure of areas subjected to significant changes, which can be due to underlying natural and human-made phenomena using sequences of Sentinel-1 SAR images collected through the IW acquisition mode.

Experimental results have been carried out by processing a series of 18 SAR images collected by the S-1A sensor from March to October 2018 over an area in the Kerala region, India, affected by massive inundations in the summer of 2018. The selected case study area is significant for the purposes of evaluating the impact of the developed method for the analyses on the effects of hydro-geological disturbances on the Earth's surface, which are more frequent now than in the past and are occurring worldwide, as triggered by present-day global climate change.

## 2. Method

This section first introduces the fundamentals of non-local mean filtering when applied to SAR images (Section 2.1). Subsequently, the developed space-time (3D) non-local mean filter is presented in Section 2.2. The theoretical performance of the proposed method is finally discussed in Section 2.3.

### 2.1. Spatial (2D) Non-Local Mean Filters for SAR Image De-Speckling

The "restoration quality" of a degraded SAR image may depend on a priori knowledge of the phenomena that caused such a degradation. Having said this, the knowledge of the noise characteristics is generally insufficient for the perfect retrieval of the suitable actual signals that are immersed within the noise; thus, the only possibility that remains is that of carrying out spatial or frequency-based filtering operations. Specifically, spatial filtering is a digital image processing operation that allows the modifying of the intensity of the involved imaged pixels. In the presence of speckle noise, SAR processing provides two distinctive approaches [95]: (i) the multi-look approach, which is applied during pre-processing, or (ii) the noise-filtering approach, which is applied during SAR processing. The conventional multi-look operation consists in computing the incoherent average of *N*-imaged SAR pixel estimates that can be extracted by considering local SAR pixels and/or splitting the image spectra into *N-independent* segments (looks). Subsequently, these *N* estimates, processed independently, are packed together in order to obtain an image where the standard deviation of the speckle is decreased by a factor $\sqrt{N}$, but at the expense of the spatial resolution, which is degraded by a factor *N*.

This work explores the potential of NLMF. In 2005, Buades et al. [83] proposed a new method in order to better de-noise images corrupted by additive Gaussian noise. To introduce the problem, let us assume we have a noisy image $v$. Let $v_f$ be the corresponding (unknown) noise-free image. Further, NL means the estimates of the value of the filtered image at the pixel of coordinates $s$ via the weighting of the averaging of the similar pixels as [55,96]:

$$\hat{v}_f(s) = \frac{\sum\limits_{p \in \Omega} w(s, p) v(p)}{\sum\limits_{p \in \Omega} w(s, p)} \tag{1}$$

where $\Omega$ is the entire image, or a sector of the image, and $w(s, p)$ is a weight that allows one to compute the grade of similarity between two patches, namely $\Upsilon(s)$ and $\Upsilon(p)$ of an image of adequate size, which are centered around the location of pixels $s$ and $p$, respectively.

Figure 1 shows a pictorial representation of the functioning principles of NLMF. The red box identifies a patch centered on the reference pixel $s$; the yellow box represents a non-local window centered on the location of the generic pixel $p$, which can span the entire image or a portion of it (e.g., the area inside the dashed blue contour line in Figure 1).

While additive Gaussian noise is present, the weights are profitably set by considering the Euclidean distance between the two patches using a filter with a Gaussian kernel, as follows:

$$w(s, p) = \exp\left[-\frac{d^2(\Upsilon(s), \Upsilon(p))}{h^2}\right] \tag{2}$$

where $h$ is a parameter that represents the variance of the adopted Gaussian filter, permitting us to modulate the strength of the NL mean filtering operation.

Several adaptations of the original NLMF have been proposed. They have all looked to deploy that method in order to obtain amplitude SAR images and InSAR products [82,84–87,97–99]. The key point is that amplitude SAR images are characterized by multiplicative speckle noise. Accordingly, the original method cannot simply be applied to SAR images. To cope with this issue properly, a few modifications have been used in order to consider both SAR image heterogeneities and the multiplicative nature of the speckle noise [100]. For instance, in [101], the image ratio was employed in order to evaluate the distance metric

between two SAR patches using adaptive filter weights based on computing the coefficient of variation (CV) of a SAR patch, which is defined as $CV = \sigma/\mu$. Where $\sigma$ and $\mu$ are the standard deviation and the average value of the image computed in an averaging window, respectively. Specifically, if $v$ is a noisy SAR image with $N \times M$ (azimuth $\times$ range) elements, the filtered SAR image is given by Equations (1) and (2) where the distance metric is assigned by:

$$
d\big(\Upsilon(s), \Upsilon(p)\big) = \frac{\sum\limits_{k \in \Psi} \exp\left(-|CV(s,k) - CV(s)|^2\right) \left| \frac{v_s(k)}{v_p(k)} + \frac{v_p(k)}{v_s(k)} \right|^2}{\sum\limits_{k \in \Psi} \exp\left(-|CV(s,k) - CV(s)|^2\right)}
\tag{3}
$$

with $CV(s,k)$ as the local CV of the generic pixel $k$ within the patch $\Upsilon(s)$, i.e., the set $\Psi$, and $CV(s)$ is the local CV of the pixel $s$. In order to adapt the filtering strength moving from homogeneous to heterogeneous regions, the authors of [96] also proposed to use the following adaptatively chosen weighting parameter $h'$ instead of $h$:

$$
h' = h \left[ 1 - \frac{1}{1 + e^{\zeta\left(\frac{CV - \sigma}{CV_{\max}}\right)}} \right]
\tag{4}
$$

where $\zeta$ is a value in the range [20,50], $CV$ is the local coefficient of variation of the image, and $CVmax$ is the maximum CV value of the whole SAR image.

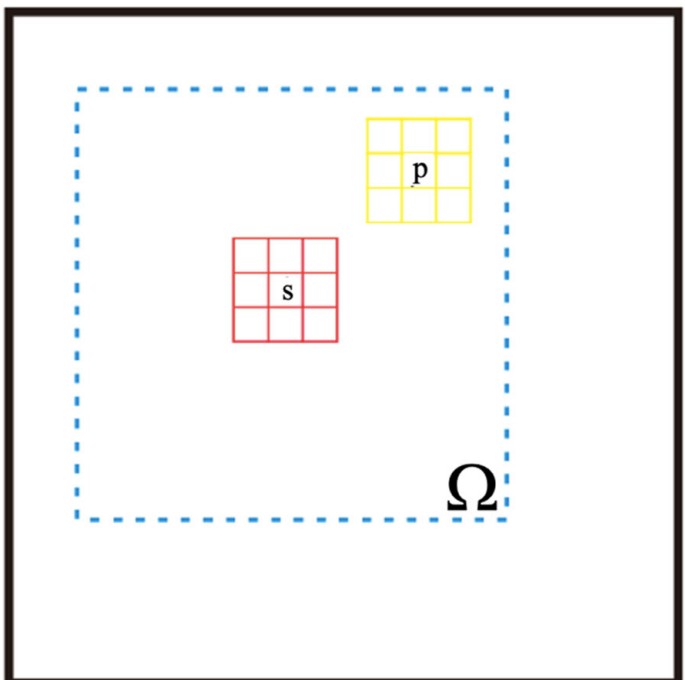

**Figure 1.** Pictorial representation of the windows used by NLMF.

### 2.2. The Developed Space-Time (3D) NLMF

Let us now assume the availability of a stack of $Q$-independent SAR single-look-complex (SLC) images $v_i$, $i = 0, \ldots, Q-1$ that are collected at ordered times $[t_0, t_1, \ldots, t_{Q-1}]$, which have preliminarily been co-registered to one reference SLC image. While de-speckling the generic $q$-th SAR image, the basic idea of the developed method is to extract similar patches not only from the $q$-th image, but from the whole set of available SAR images, thus increasing the total number of looks and enhancing the quality of filtered SAR images. Mathematically, in this case, Equation (1) modifies as follows:

$$\hat{v}_{q,f}(s) = \frac{\sum\limits_{i=0}^{Q-1} \sum\limits_{p \in \Omega} w_i(s_q, p) v_i(p)}{\sum\limits_{i=0}^{Q-1} \sum\limits_{p \in \Omega} w_i(s_q, p)} \qquad (5)$$

where the weights $w_i(s_q, p)$ $\forall i = 0, \ldots, Q-1$ are computed by considering the pairs of SAR patches extracted from the same or a different SAR image, thus extending to the three-dimensional (3D) case that has already been developed for the two-dimensional (spatial) case. Figure 2 pictorially describes how the developed 3D NLMF method works. Further, the $Q$ layers represent the available SAR images. When the noise in a single SAR image is filtered out, patches are extracted from the same (see the example of layer 0 in this picture) or from different SAR images. The green lines pictorially represent the moving windows that allow us to dynamically identify the reference and the adaptive patches that cover the whole picture or even a sector of every single layer into the stack of SAR images. The straightforward method proposed here was preliminarily tested on a selected test case, and the relevant results were first presented in a conference paper [102]. With respect to it, the present investigation: (i) further explores the validity of the developed 3D method when adaptatively chosen weights (Equations (3) and (4)) are used for the purposes of the analysis of heterogenous scenes; (ii) it adopts synthetic indicators to test the potential and the limitations of the method (see Section 2.3), and (iii) investigates implications for the temporal multi-looking.

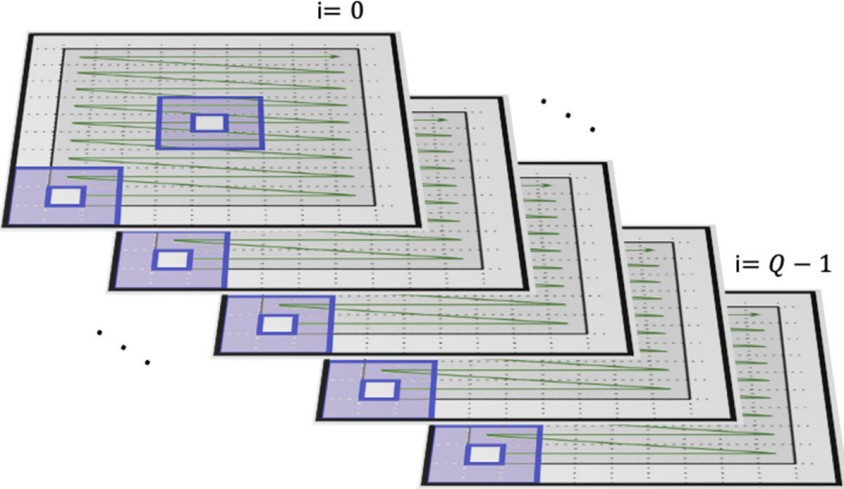

**Figure 2.** Pictorial representation of the windows used by the developed 3D NLMF method.

### 2.3. Key Performance of the Developed 3D NLMF De-Speckling Method

In order to evaluate the de-noising performance of the developed NLMF method, different key estimators can be used, such as: the equivalent number of looks (ENL), the edge-preserving index (EPI), and the peak signal-to-noise ratio (PSNR), which are subsequently shortly summarized [103].

The ENL factor determines the grade of smoothness of an image; it is defined as:

$$ENL = \left(\frac{\mu}{\sigma}\right)^2 \qquad (6)$$

where $\mu$ and $\sigma$ are the (local) mean and standard deviation of the filtered image.

Conversely, EPI is used to evaluate the capability of correct retention of the edges in an image; it is defined as:

$$EPI = \frac{\sum\limits_{P\in\Xi} \left|\frac{\partial v_f}{\partial x}(P)\right| + \left|\frac{\partial v_f}{\partial y}(P)\right|}{\sum\limits_{P\in\Xi} \left|\frac{\partial v}{\partial x}(P)\right| + \left|\frac{\partial v}{\partial y}(P)\right|} \qquad (7)$$

where $v_f$ and $v$ are the filtered and the un-filtered images, respectively, $\Xi$ is the entire image or a (local) sector of it and $x$ an $y$ are the generic 2D spatial dimensions of the considered image.

Finally, PSNR is defined over a patch of the image as follows:

$$PSNR = 10\log_{10}\left[\frac{(\max(v))^2}{mse\left(v, v_f\right)}\right] \qquad (8)$$

where $mse\left(v, v_f\right)$ and $\max(v)$ are the mean square of the difference between the filtered and the unfiltered image and the maximum value of the unfiltered image, respectively: The larger the PSNR, the better the de-noising effect.

## 3. Material

A dataset consisting of 18 SLC SAR images collected from 6 March to 20 October 2018—through the terrain observation with progressive scans (TOPSAR) mode with Sentinel-1A sensor (VV polarization)—was used in order to test the performance of the developed 3D NLMF method. Table 1 lists the SAR images used for the experiments shown in Section 4.

**Table 1.** List of S-1 acquisitions (VV polarization, descending orbits, and Path 165) used for the experiments carried out over the Kerala (India) area.

| Year | Month | Day |
|------|-------|-----|
| 2018 | 03 | 06 |
| 2018 | 03 | 18 |
| 2018 | 03 | 30 |
| 2018 | 04 | 11 |
| 2018 | 04 | 23 |
| 2018 | 05 | 05 |
| 2018 | 05 | 17 |
| 2018 | 05 | 29 |
| 2018 | 06 | 10 |
| 2018 | 06 | 22 |
| 2018 | 07 | 04 |
| 2018 | 07 | 16 |
| 2018 | 07 | 28 |
| 2018 | 08 | 09 |
| 2018 | 08 | 21 |
| 2018 | 09 | 02 |
| 2018 | 09 | 14 |
| 2018 | 09 | 26 |
| 2018 | 10 | 08 |

The selected case study is relevant to the Kerala region, India, as shown in Figure 3. The area was subjected to severe floods during the summer of 2018 [104,105]. Heavy monsoon rainfalls resulted in dams filling their operational capacity, causing the flood of low-lying regions, with over 500 dead people, hundreds of villages destroyed, and millions of USD in economic losses. In particular, the investigations presented in Section 4 refer to the inundations that occurred in the Thrissur District.

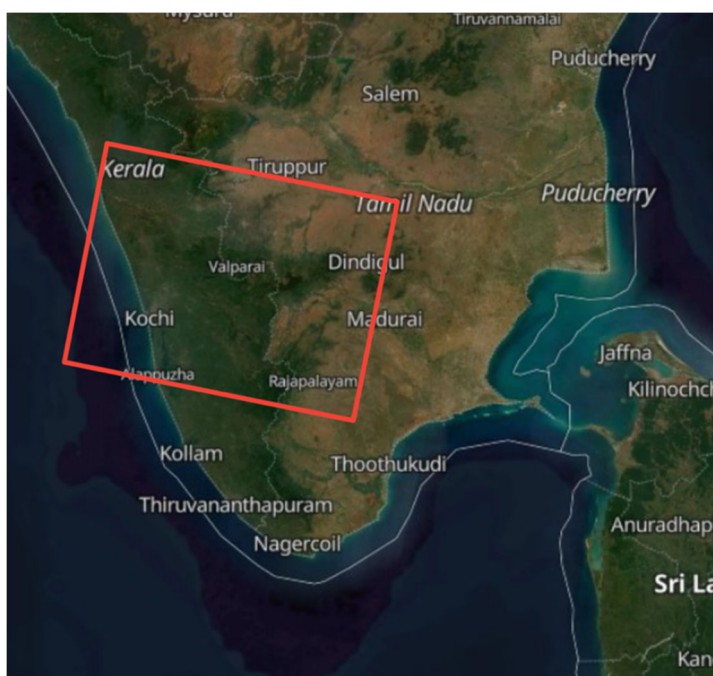

**Figure 3.** Case study area of the Kerala region, India. The red square box identifies the footprint of the used IW S1-A data collected from the descending orbits (Path 165 and Frame 558).

## 4. Experimental Results

The available SAR images were preliminarily radiometrically calibrated through the Sentinel Application Platform (SNAP) [106] in order to extract, from the digital data maps of every single SAR image, the radar backscattering (sigma naught) values. It is worth remarking that, in order to evaluate the performance of the developed NLMF algorithm, the author of this investigation operated primarily in radar (azimuth × range) coordinates. Figure 4a,b show the maps of (unfiltered) sigma naught images computed over the Thrissur District on the two acquisition dates: 6 March 2018 (pre-) and 22 June 2018 (post-flood). The images were converted in log-scale (dB) for a suitable representation of the computed backscattered signal values. Figure 4c also shows the ratio image (post/pre), which corresponds to the difference between the images in Figure 4a,b. Maps are shown after a multi-look operation (with 20 pixels in range and 4 pixels in the azimuth directions, respectively) has been performed on the SAR images, thereby possessing images to show with almost squared pixels.

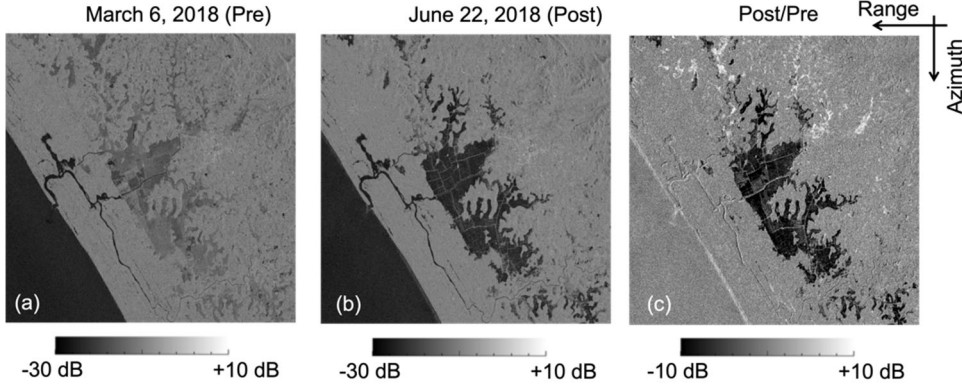

**Figure 4.** Maps of the sigma naught values (dB) computed on the unfiltered SAR images acquired before (**a**) and after (**b**) the occurrence of inundations in the Kerala region. The map in (**c**) shows the ratio image (i.e., the difference between (**b**) and (**a**)).

For the readers' convenience, Figure 5 shows the ratio image map (dB scale) superimposed on Google Earth in order to better identify the geographical location of the investigated region.

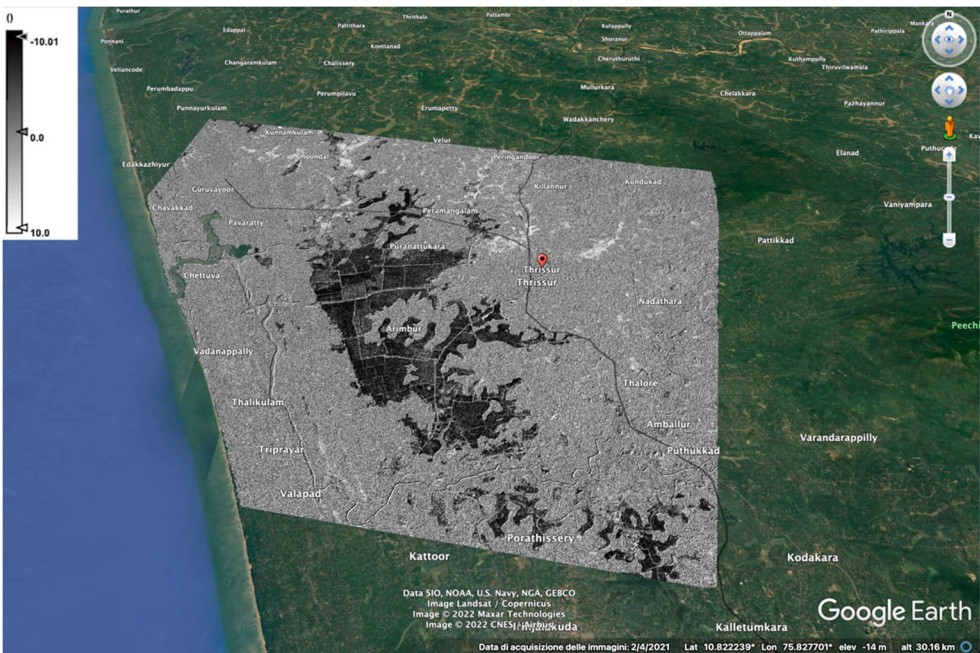

**Figure 5.** Case study area of the Kerala region, India, superimposed on the map of flooded regions.

The corresponding results obtained using the 2D NLMF method described in Section 2.1 are shown in Figure 6a–c.

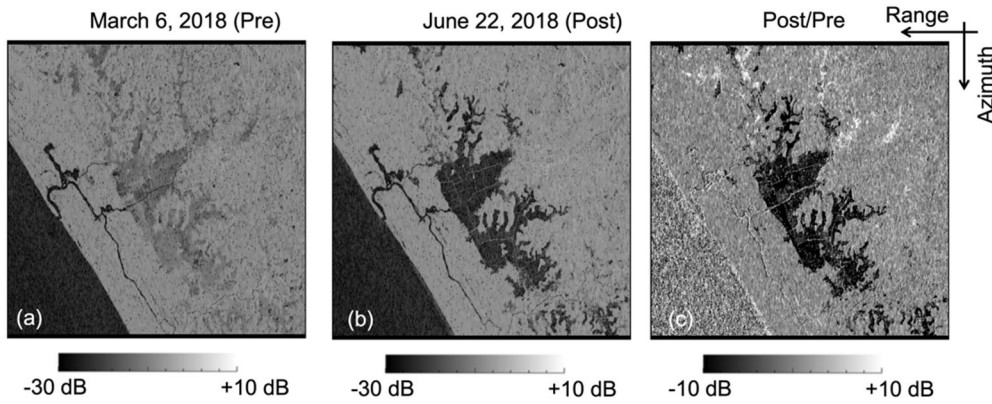

**Figure 6.** These are the same graphs as shown in Figure 4 but using the filtered SAR images, computed by independently applying the non-local mean filter as described in Section 2.1 to every SAR image (2D case). (**a**–**b**) Maps of the sigma naught values (dB) computed before and after the main flood events, and (**c**) is the ratio image between (**a**) and (**b**) in log scale.

Figure 7 shows the equivalent results obtained by applying the developed 3D NLMF method described in Section 2.2.

Note that the maps shown in Figures 4, 6 and 7 were obtained after applying a multilook operation (with 20 pixels in range and 4 pixels in the azimuth directions, respectively) on the single-look SAR images/products, thereby resulting in possessing images with almost squared pixels. It is also worth remarking that non-local mean filter operations had been carried out considering "similarity windows" with a dimension of (20 × 20 pixels). The "similar" patches were searched over the 100 × 100 pixels centered on the analyzed, generic SAR pixel in order to reduce the overall algorithm computation time. Of course,

the proposed method is naturally prone to a parallel implementation on multi-core/multi-node computing architectures. As a further remark, it is worth highlighting that no other preliminarily or post-processing noise-filtering algorithms were applied to available SAR images. It was explicitly intended to elucidate the potential of the developed 3D noise-filtering method. Nonetheless, the joint application of additional pre-processing noise-filtering procedures were, for instance, using [69,70]. In addition, the subsequent application of the developed de-speckling method could enhance expected performance while preserving spatial details.

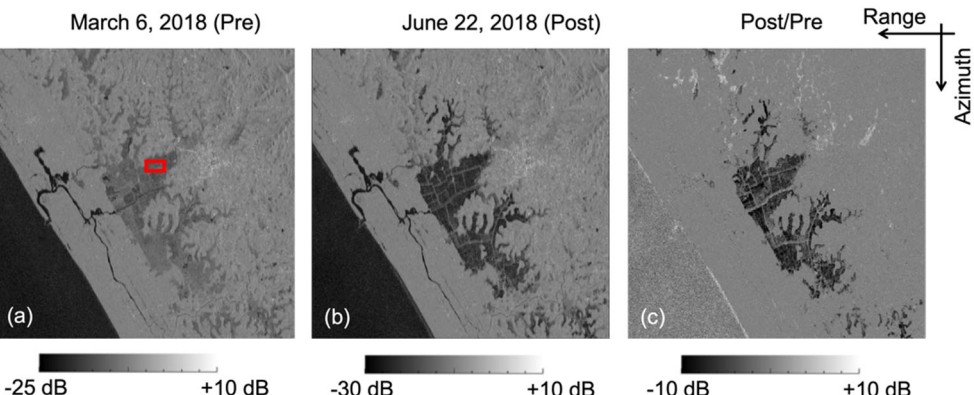

**Figure 7.** These are the same graphs as shown in Figure 4 but using the 3D NLMF filtered SAR images. (**a**,**b**) Maps of the sigma naught values (dB) computed before and after the main flood events, and (**c**) is the ratio image between (**a**) and (**b**) in log scale.

## 5. Key Performance of the Proposed Method

In order to quantify the performance of the 3D NLMF method, first, the NLMF method was applied to 2D images separately, as described in Section 2.1. Subsequently, the 3D method explored in Section 2.2 was used for the sequence of available SAR images.

The selected case study area represents an excellent example of a heterogeneous scene. In order to evaluate the performance and limitations of the presented 3D filtering method, the subsequent analyses focus on the evaluation of crucial results obtained by considering different values for the standard deviation (i.e., $h'$) of the Gaussian kernel shown in Equations (3) and (4). Moreover, the selected value for the $\xi$ parameter was set to $\xi = 50$. Then, the value of the constant parameter $h$ was varied accordingly. For every pool of parameters, the values of averaged ENL, EPI, and PSLR were calculated in the context of considering, as a test, the region highlighted with the red rectangle drawn in Figure 7a. Figure 8 shows the zoomed view of the 3D NLMF de-speckled SAR image (represented in a linear scale) relevant to the SAR acquisition data collected on 22 June 2018, as obtained considering different values of the parameter $h$. The ENL, EPI, and PSLR were initially computed using a moving averaging window of $20 \times 20$ pixels and then averaging them over the selected area.

As a further experiment, the size of the "similar patches" was varied, considering the cases of the $10 \times 10$ and $30 \times 30$ SAR pixels. Figure 9 shows the 3D filtered maps of 22 June 2018, obtained with different patches sizes and Gaussian filter standard deviations. The results show that small patches sizes (i.e., $10 \times 10$) are responsible for a dramatic smoothing of the filtered SAR image, as shown in Figure 9b, due to lack of information in the estimation of the patches distance. Conversely, the noise-filtering performance improvements in terms of enhanced ENL values with larger patches sizes is paid in terms of higher computation times.

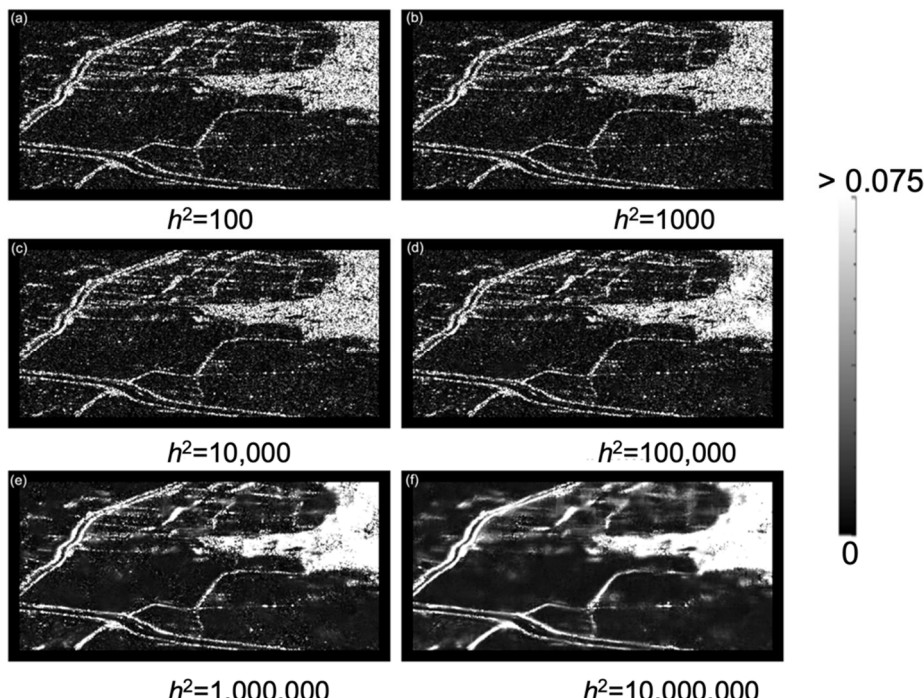

**Figure 8.** The zoomed view of a selected area of the 3D NLMF images. (**a**–**f**) are the results obtained by considering $h^2$ = 100 (**a**), $h^2$ = 1000 (**b**), $h^2$ = 10,000 (**c**), $h^2$ = 100,000 (**d**), $h^2$ = 1,000,000 (**e**) and $h^2$ = 10,000,000 (**f**).

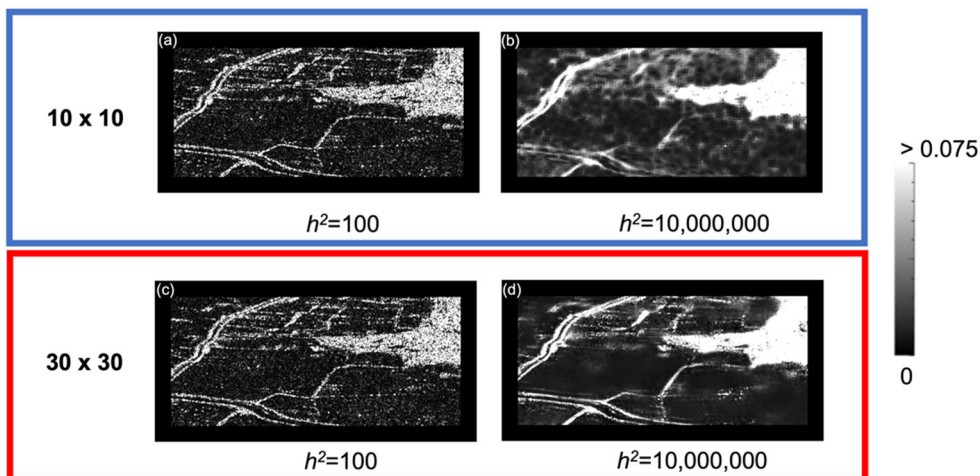

**Figure 9.** The zoomed view of a selected area of the 3D NLMF images. (**a**,**b**) are the results obtained by considering $h^2$ = 100 (**a**), $h^2$ = 10,000,000 (**b**) using patches size of 10 × 10. (**c**,**d**) are same as (**a**,**b**) but with patch sizes of the 30 × 30 SAR pixels.

Figure 10 shows the plots of ENL, EPI, and PSLR vs. the variance of the Gaussian kernel ($h^2$), in the logarithmic scale, which are for the two separate cases in which the 2D NLMF and 3D NLMF procedures are applied, when considering the patch sizes of the 20 × 20 SAR pixels.

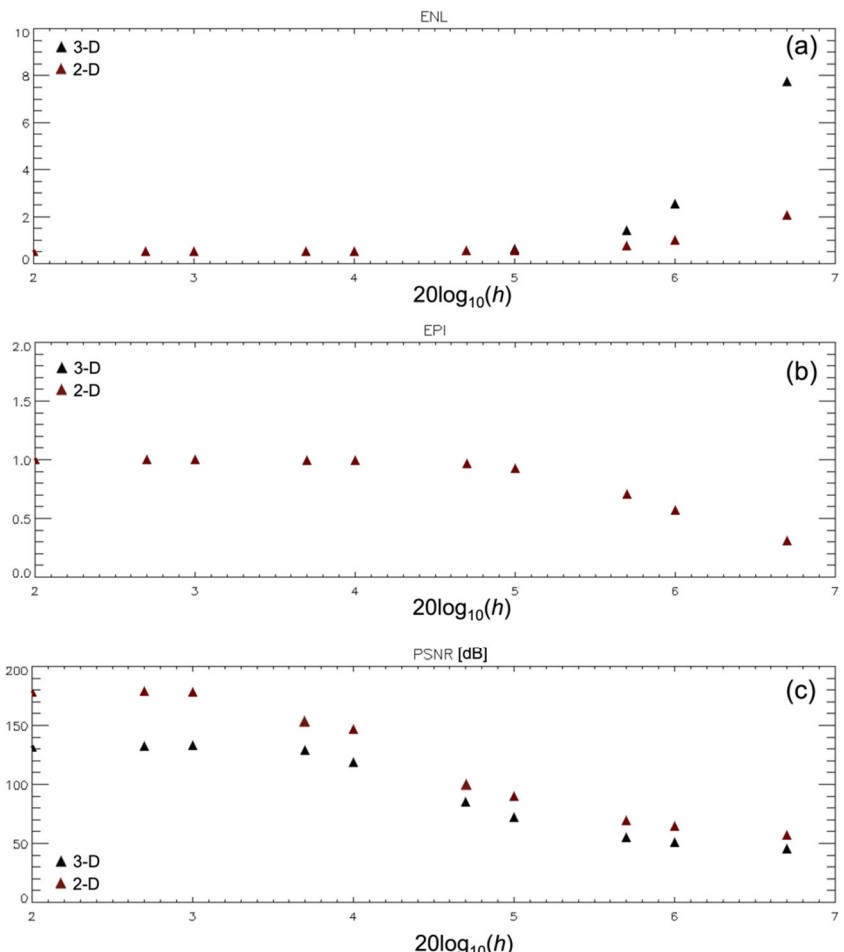

**Figure 10.** A plot of the ENL (**a**), EPI (**b**), and PSLR (**c**) vs. the standard deviation of the used Gaussian filter. The 3D results are represented with red triangles and 2D with black triangles. The results refer to an averaging patch size of $20 \times 20$ pixels.

The analysis of Figure 10 reveals that the correct choice of the Gaussian kernel must necessarily result from a compromise between noise reduction and feature extraction preservation. More specifically, small values of $h$ correspond to high values of PSLR and EPI. In contrast, higher values of $h$ significantly ameliorate ENL at the expense of a reduced capacity to identify small features and preserve edges. EPI is practically the same for both the 2D and 3D cases. ENL improvements of the 3D case with respect to the 2D one, using the same parameters of the Gaussian Filter in Equations (2)–(4), appear predominant with values of $h^2$ larger than $5 \times 10^5$. The results demonstrate a balance between the different aspects that must be adopted in order to tune the 3D NLMF. A strategy that may be assumed in order to identify the correct value of $h^2$ to be used is, for instance, found in determining the elbow point of the curve, as shown in Figure 10a.

The obtained noise-filtered images can be beneficial for the purposes of applying conventional and multi-temporal change detection (CD) methods, considering several heterogeneous contexts and natural/anthropogenic risk conditions. This work only intends to provide readers with information concerning the newly developed 3D NMLF algorithm and its performance. However, further analyses of the identified CD maps are worth subsequent investigations. Moreover, exploiting extensive archives of processed SAR data is mandatory in order to access the potential of the developed method in a fully operational context.

## 6. Conclusions

This paper investigated the potential of 3D non-local mean filters (NLMF) for the purposes of de-speckling SAR images, particularly in change detection applications. The theory of NLMF was briefly introduced and a specific implementation was considered in order to translate the problem from the 2D to the 3D domain. The proposed methodology was initially tested on a region in Kerala, India, which was a region that was severely affected by disastrous flooding inundations in 2018. This method can be used to analyze several heterogenous catastrophic conditions by the improving of the spatial resolution of data products while profiting from the time redundancy of information gathered in a sequence of multi-temporal SAR images. The noise-filtering operation was conducted adaptatively in order to preserve the capability to recover even the tiny amplitude variations from one image to another. This method was implemented considering SAR applications involving sets of amplitude (and calibrated) SAR images. However, the same concept can also be adapted to better filter out the noise from a sequence of multi-temporal SAR interferograms, operating directly in the 3D space/time domain, in conjunction with other methods (see for instance [17,107–109]). Furthermore, it must be noted that this was recently proposed in the literature for ground deformation retrieval through InSAR technology [110,111].

**Funding:** This research received no external funding.

**Data Availability Statement:** The data presented in this study are available on request from the corresponding author.

**Acknowledgments:** The author thanks ESA, who provided the Sentinel-1 SAR data, and Vincenzo Sileo, who preliminarily contributed to the processing of the sigma naught maps within the framework of his master's degree thesis. Additionally, the author would like to express his gratitude to Mariarosaria Manzo for the longest and happiest hours spent together in twenty years, and he hopes to see her again elsewhere.

**Conflicts of Interest:** The author declares no conflict of interest.

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
