# Peer review of "A 3D Space-Time Non-Local Mean Filter (NLMF) for Land Changes Retrieval with Synthetic Aperture Radar Images"

_remotesensing, doi:10.3390/rs14235933_

Round 1

Reviewer 1 Report

Dear Author,

The manuscript is well-written and the topic is of interest to the readers. I have the following comments;

1. The following paper from the author is not cited, although it seems like the presented study is an extension of it:

A. Pepe, "Use of Multi-Temporal SAR Non-Local Mean Filtering Operations for Change Detection Analyses," 2022 IEEE 21st Mediterranean Electrotechnical Conference (MELECON), 2022, pp. 616-620, doi: 10.1109/MELECON53508.2022.9842906.

 The differences between the manuscript and the conference paper need to be explained.

2. The captions of Fig 1 and Fig 2 are very long. I recommend to move parts of the information to the main text.

3. The results and discussions are not sufficiently given. The performance and the advantages of the proposed method are not clearly understandable in the manuscript.

Author Response

Comment:

The manuscript is well-written and the topic is of interest to the readers. I have the following comments; 

Reply:

I thank the reviewer for her/his comments on the paper.

Comment:

  1. The following paper from the author is not cited, although it seems like the presented study is an extension of it:
  2. Pepe, "Use of Multi-Temporal SAR Non-Local Mean Filtering Operations for Change Detection Analyses," 2022 IEEE 21st Mediterranean Electrotechnical Conference (MELECON), 2022, pp. 616-620, doi: 10.1109/MELECON53508.2022.9842906.

 The differences between the manuscript and the conference paper need to be explained.

Reply:  

I thank the reviewer for this comment. The previous conference paper has been cited, and a short explanation of the differences between the conference and the full paper is introduced.

Comment:

  1. The captions of Fig 1 and Fig 2 are very long. I recommend to move parts of the information to the main text.

Reply:

 I have followed the raise reviewer’s recommendations.

Comment:

  1. The results and discussions are not sufficiently given. The performance and the advantages of the proposed method are not clearly understandable in the manuscript.

Reply:

 I would like to thank the reviewer for this comment. In the revised manuscript, the performance, advantages but also limitations of the proposed method are better discussed.

Reviewer 2 Report

The paper considers the extension to 3D of a despeckling algorithm. There is no consideration to any prior change detection. Results are given on a flood in Kerala.

To consider the time axis equivalent to the three space axes is an important assumption, that is in general not acceptable. So, I regret but it is my opinion that this paper has to be rejected, as change detection analyses have to precede any smoothing that will, by definition, reduce the change.

It is my opinion that this paper, as it is,  would mislead the readers.

I also add references to just a few of the numerous papers that  refer to change detection. Many other can be found referring to floods, as seen with SAR.

SAR Image Despeckling by Selective 3D Filtering of Multiple Compressive Reconstructed Images By Mahboob Iqbal Jie Chen Wei Yang Pengbo Wang Bing Sun Progress In Electromagnetics Research, Vol. 134, 209-226, 2013doi:10.2528/PIER12091504

Y, Murali Mohan Babu & Subramanyam, M V & M.N, Giri Prasad. (2015). A Modified BM3D Algorithm for SAR Image Despeckling. Procedia Computer Science. 70. 69-75. 10.1016/j.procs.2015.10.038.

Ma X, Wu P. Multitemporal SAR Image Despeckling Based on a Scattering Covariance Matrix of Image Patch. Sensors (Basel). 2019 Jul 11;19(14):3057. doi: 10.3390/s19143057. PMID: 31373333; PMCID: PMC6678814.

Google Earth Engine; Detecting Changes in Sentinel-1 Imagery, Python Engine.

Author Response

Comment:

The paper considers the extension to 3D of a despeckling algorithm. There is no consideration to any prior change detection. Results are given on a flood in Kerala.

To consider the time axis equivalent to the three space axes is an important assumption, that is in general not acceptable. So, I regret but it is my opinion that this paper has to be rejected, as change detection analyses have to precede any smoothing that will, by definition, reduce the change. 

It is my opinion that this paper, as it is, would mislead the readers.

Reply:

I understand the observation of the reviewer. I have found other papers in the literature where the nature of temporal axes has been considered. Distance metrics used by NLMF methods can naturally adapt the filter weight moving from space to time axes and take into account the scene's heterogeneity. The revised paper has addressed this issue. I do not want to comfute the reviewer’s ideas. However, I do not agree that in the literature noise filtering is not adopted in change detection analyses because it smooths SAR signals. Everything depends on the scale of the analyzed changed areas. In this work, we are searching for those significant changes that can be emphasized when noise is reduced, at the expense of minor changes that fall within the error bar of noise.

Comment:

I also add references to just a few of the numerous papers that refer to change detection. Many other can be found referring to floods, as seen with SAR.

SAR Image Despeckling by Selective 3D Filtering of Multiple Compressive Reconstructed Images By Mahboob Iqbal Jie Chen Wei Yang Pengbo Wang Bing Sun Progress In Electromagnetics Research, Vol. 134, 209-226, 2013doi:10.2528/PIER12091504

Y, Murali Mohan Babu & Subramanyam, M V & M.N, Giri Prasad. (2015). A Modified BM3D Algorithm for SAR Image Despeckling. Procedia Computer Science. 70. 69-75. 10.1016/j.procs.2015.10.038.

Ma X, Wu P. Multitemporal SAR Image Despeckling Based on a Scattering Covariance Matrix of Image Patch. Sensors (Basel). 2019 Jul 11;19(14):3057. doi: 10.3390/s19143057. PMID: 31373333; PMCID: PMC6678814. Google Earth Engine; Detecting Changes in Sentinel-1 Imagery, Python Engine.

Reply:

I thank the reviewer for this valuable observation. Following her/his suggestions, I have added some references to the work and clarified better the aim of my investigation, which can be seen as a “note” for further readers and scholars to test simultaneous 3-D methods for NLMF. I hope that the additional clarifications are sufficient to define the pros and the limitations of the presented methods.

Round 2

Reviewer 1 Report

Thanks for the revisions.

Author Response

thanks

Reviewer 2 Report

The paper is ok for me now. It would be useful to complement Fig. 8 with a figure where the size of the local 3D window is represented with a proper color scale, again as a function of h. Maybe also add another one, this time   representing the 3D shape of some of the widest ones. 

Author Response

Dear reviewer, thanks for the valuable received comments during the whole revision process. Considering the final comment, I have included in the revised manuscript a short analysis related to the impact of the patches size on the 3-D noise-filtered images and added a new figure, see Figure 9.